# SARS-CoV-2 Droplet and Airborne Transmission Heterogeneity

**DOI:** 10.3390/jcm11092607

**Published:** 2022-05-06

**Authors:** Marta Baselga, Antonio Güemes, Juan J. Alba, Alberto J. Schuhmacher

**Affiliations:** 1Institute for Health Research Aragon (IIS Aragón), 50009 Zaragoza, Spain; mbaselga@iisaragon.es (M.B.); aguemes@unizar.es (A.G.); jjalba@unizar.es (J.J.A.); 2Department of Surgery, University of Zaragoza, 50009 Zaragoza, Spain; 3Department of Mechanical Engineering, University of Zaragoza, 50018 Zaragoza, Spain; 4Fundación Agencia Aragonesa para la Investigación y el Desarrollo (ARAID), 50018 Zaragoza, Spain

**Keywords:** COVID-19, bioaerosols, infectious diseases, cough, airborne, superspreaders, transmission heterogeneity

## Abstract

The spread dynamics of the SARS-CoV-2 virus have not yet been fully understood after two years of the pandemic. The virus’s global spread represented a unique scenario for advancing infectious disease research. Consequently, mechanistic epidemiological theories were quickly dismissed, and more attention was paid to other approaches that considered heterogeneity in the spread. One of the most critical advances in aerial pathogens transmission was the global acceptance of the airborne model, where the airway is presented as the epicenter of the spread of the disease. Although the aerodynamics and persistence of the SARS-CoV-2 virus in the air have been extensively studied, the actual probability of contagion is still unknown. In this work, the individual heterogeneity in the transmission of 22 patients infected with COVID-19 was analyzed by close contact (cough samples) and air (environmental samples). Viral RNA was detected in 2/19 cough samples from patient subgroups, with a mean Ct (Cycle Threshold in Quantitative Polymerase Chain Reaction analysis) of 25.7 ± 7.0. Nevertheless, viral RNA was only detected in air samples from 1/8 patients, with an average Ct of 25.0 ± 4.0. Viral load in cough samples ranged from 7.3 × 10^5^ to 8.7 × 10^8^ copies/mL among patients, while concentrations between 1.1–4.8 copies/m^3^ were found in air, consistent with other reports in the literature. In patients undergoing follow-up, no viral load was found (neither in coughs nor in the air) after the third day of symptoms, which could help define quarantine periods in infected individuals. In addition, it was found that the patient’s Ct should not be considered an indicator of infectiousness, since it could not be correlated with the viral load disseminated. The results of this work are in line with proposed hypotheses of superspreaders, which can attribute part of the heterogeneity of the spread to the oversized emission of a small percentage of infected people.

## 1. Introduction

After more than two years of the COVID-19 pandemic, the role of the different transmission routes in the dynamics of the spread of the SARS-CoV-2 virus are still unknown. Human–human transmission has been described from direct respiratory spread, where the symptomatic or asymptomatic patient expels contaminated particles in respiratory events, and, on the other hand, indirect spread or via fomites, where the transmission is due to contact with contaminated surfaces. Regarding direct dissemination, fomites are objects that can be contaminated with pathogenic microorganisms and serve as transmission vehicles [1,2]. Variable SARS-CoV-2 survival depends on the surface material; it remains active for up to 72 h on plastic and stainless steel, up to 24 h on cardboard, and less than 4 h on copper [3]. On polymeric surfaces and skin, the Alpha, Beta, Delta, and Omicron variants exhibited survival twice as high as the ancestral strain, reaching a persistence of more than 16 h on the skin [4].

It is possible to differentiate between the droplet model and the bioaerosol model regarding indirect dissemination. Droplets and bioaerosols differ primarily in their aerodynamic diameter and properties. Large droplets predominate in close contact, but they are not capable of infecting individuals at a distance, since they settle rapidly due to gravitational effects. However, the aerodynamic diameter of aerosols allows them to be transmitted over time and distance [5], as represented in Figure 1. This interaction phenomenon between gravity and evaporation, depending on the size of the drop, was described in 1955 by Wells [6]. Suspension time can be determined from Stokes’ Law, which describes an inverse relationship between particle size and deposition rate. For example, a 5 µm aerosol takes about 33 min to settle on the ground, while a 1 µm aerosol remains suspended in the air for more than 12 h [7]. Regarding this pandemic, the scientific community has redefined the concept of bioaerosol, extending its consideration to airborne particles smaller than 100 µm, based on evidence and common factors related to the aerodynamics of the particles. Nevertheless, it is necessary to consider that both large droplets and bioaerosols are dynamic, since they can modify their morphology and physical–chemical properties, depending on environmental parameters. In this way, as we will discuss later, small droplets can become aerosols and, in turn, larger aerosols can migrate towards smaller aerosols, modifying their aerial dynamics. Fine particles pose greater challenges when it comes to preventing potential contagion: on the one hand, because they correspond to the size range where the retention efficiency of filters is lower (they typically have a minimum efficiency around 300 nm [8,9]); on the other, because they can remain suspended indefinitely and reach greater distances from their emitter.

The viruses are transported through small secretions of saliva and mucosa in infected people, inadvertently expelled through the nose and mouth. It is possible to differentiate between bronchiolar, laryngeal, and oral bioaerosols, depending on the generation mechanism and anatomical region where they originate. Bronchiolar particles are predominantly attributed to normal breathing and are associated with the rupture of the fluid film in the bronchioles by shear forces [10]. The vibration of the vocal cords generates laryngeal secretions during speech and vocalization [11]. Instead, oral particles where droplets are larger than 100 µm are predominantly produced from saliva in the oral cavity. Their emission rate and velocity depend on the effort during the vocalization event [11]. Other study suggests a dependence between the rate of aerosol emission and the amplitude of vocalization and, furthermore, points to the existence of independent “superspreading” events [12]. These bioaerosols are predominantly composed of ions (predominantly Na^+^, Cl^−^, Ca^2+^, and Mg^2+^), organic and inorganic particles and glycoproteins, mainly albumin, mucins, cholesterol and pulmonary surfactant proteins, such as DPPG and DPPC [13]. They can also include, in infected individuals, viral or bacterial pathogens. This is especially relevant during infections that present a high viral load in the upper respiratory tract due to the anatomical proximity to the ‘escape routes’, as is the case for COVID-19 [3,14,15].

The airborne model of COVID-19 disease transmission has been demonstrated [16] among small animals [17,18,19], from viral superspreading events [20], in long-distance transmission scenarios where infected individuals do not come into direct contact [21], by transmission in asymptomatic individuals [22], and by the prevalence of spread in closed spaces [23]. Despite the alerts by scientific groups since April 2020, this route of contagion was dismissed, and greater attention was paid to fomites and contagion by droplets, following the classic models of transmission of respiratory diseases for the COVID-19 pandemic. Consequently, there was controversy about whether asymptomatic infected individuals could be transmitters of SARS-CoV-2, which was an obstacle during epidemiological management. The global acceptance of the spread of COVID-19 by aerosols has modified the preventive approach, including new measures to reduce the risk of contagion. In April 2021, this route of infection was accepted by the WHO as one of the main ones [24].

The size of the SARS-CoV-2 virion varies between 70 and 90 nm [25,26], and a mean virus concentration in sputum of 7.0 × 10^6^ copies/mL and a maximum of 2.4 × 10^9^ copies/mL has been reported [27]. Consequently, the viral load occupies 2.1 × 10^−6^ % of the bioaerosol on average. With this value, Lee [28] estimated a theoretical minimum and initial aerosol size of 4.7 µm to contain SARS-CoV-2. However, experimental bioaerosol sampling studies suggest the presence of the virus in smaller particle sizes (even <0.25 µm). Liu et al. [29] were pioneers in experimentally investigating the aerodynamic nature of the SARS-CoV-2 pathogen by quantifying viral RNA from aerosols in different hospital areas in Wuhan (China) during one of its most severe outbreaks. In their work, they determined the presence of viruses predominantly in two size ranges: in the submicrometric region (0.25–1.0 µm) as well as in the supermicrometric region (>2.5 µm). Since then, numerous scientific efforts have aimed at characterizing and understanding the dynamics of the spread of COVID-19 associated with aerosols. However, the role of airborne transmission for the SARS-CoV-2 virus and the risk of contagion that they might represent have not yet been well described.

The modal distributions of aerosols acquire great relevance in the context of disease transmission. They determine the aerodynamic characteristics and their deposition dynamics, as well as the variability in the viral colonization model, depending on the depth of the respiratory tract [30]. In addition to influencing the modal distribution of aerosols, the nature of the activity carried out will modify parameters associated with the kinetics of the particles. Chao et al. [31] pointed to a mean velocity of 11.7 m/s during coughing and 3.9 m/s when speaking. Their experimental work described a geometric mean diameter of 13.5 µm when coughing and 16.0 µm when speaking, with an estimated concentration of 2.4–5.2 cm^−3^ in coughs and 0.4 × 10^−3^–0.2 cm^−3^ in speaking. The disseminating capacity of the individual is noticeably modified if the individual is speaking or coughing. Specifically, it has been described that during a one-minute conversation, more than 1.0 × 10^3^ aerosols can be disseminated [32], and an individual expels around 7.2 × 10^3^ particles per liter of exhaled air [33,34], while coughing occurs sporadically, implying a likely increased release of aerosols during breathing and speaking [35]. Subsequently, other experimental works have demonstrated, with greater or lesser success, the presence of SARS-CoV-2 viral RNA in aerosols using various sampling methods, such as solid impactors [36,37,38], cyclones [39,40,41,42,43,44,45,46,47,48,49], impingers [50,51], gelatin filters [39,44,52], particulate filters [53,54,55,56], and condensation systems [38,57,58]. However, other investigations have failed to recover detectable RNA concentrations [59,60,61,62,63,64,65,66].

Numerous factors can influence the airborne transmission of pathogens, both at the dynamics of propagation and the virus survival (or persistence). The size of exhaled bioaerosols evolves due to evaporation, coagulation, and deposition, directly affecting their air suspension time and persistence [35]. Consequently, the size distribution of the airborne concentration of aerosols will vary with time, since larger diameter particles settle faster [67]. However, other extrinsic factors, such as ambient airflows, affect both the airborne suspension time and the distance traveled by the aerosol [68]. At the level of virus survival, the presence of ions affects droplet evaporation. Therefore, the dynamics change, while sputum organic compounds are insignificant due to the low molar fraction they represent [14]. By reducing the aqueous component of the aerosol, aerosols are subject to changes in morphology, viscosity, and pH, among others, modifying the microenvironment of the virus and, therefore, reducing its persistence [69]. Specifically, low relative humidity induces evaporation, a decrease in pH and, with it, conformational changes in the proteins on its surface, making the virus a less infectious pathogen [70]. In addition, the size of the aerosol decreases proportionally until it crystallizes, significantly reducing its size in environments with very low relative humidity. In the opposite case, the aerosol tends to adsorb moisture and increase its size at high relative humidity [71]. If the relative humidity is below 80%, respiratory aerosols reach a final diameter of 20 to 40% of their original size [72]. In the case of SARS-CoV-2, it is suggested that the optimal relative humidity for minimizing the spread of this virus is between 40 and 60% [73]. Initially, Fears et al. [74] determined a greater dynamic efficiency of SARS-CoV-2 than SARS-CoV and MERS-CoV, pointing to the persistence of infectivity and virion integrity up to 16 h in aerosols (1.0—3.0 µm). Van Doremalen et al. [3] suggested a similar half-life between SARS-CoV-2 (0.6–2.6 h; 1.1 h on average) and SARS-CoV (0.8–2.4 h; 1.2 h on average). They pointed out that the discrepancies in the epidemiological characteristics could be associated with high viral loads in the upper respiratory tract and transmission of the virus in asymptomatic patients. Smither et al. [75] reported a lifespan of SARS-CoV-2 between 30 and 177 min in aerosols between 1.0 and 3.0 µm, under different conditions of relative humidity (RH), and reported a decay rate between 0.4 and 2.3%/min. Schuit et al. [76] reported independence between the decay rate and the RH of SARS-CoV-2, attributing the loss of infectivity to the effect of sunlight and the aerosol suspension medium. Aligned with the observations of Schuit et al., other authors [77,78] reported similar conclusions using other viruses. There are considerable discrepancies in the literature about viral persistence in the air. A recent study in a preprint published by Oswin et al. [79] showed that 54% of bioaerosols (5–10 µm) loaded with SARS-CoV-2 lose their infective capacity during the first 5 s at low RH (40%). Within 5 min, persistent viruses lose about 19% of their infectivity. At high RH (90%), infectivity falls by 48% progressively during the first 5 min. At 20 min, a 90% loss of infectivity has been found using different variants of SARS-CoV-2. However, these authors replaced the human mucosal base in the aerosols with a Serum with a different composition, so their direct extrapolation is limited.

To date, most environmental sampling has been carried out in hospital settings, which limits the detection of viral RNA due to constant air renewal (~12 ACH) [60,61], which implies a complete air renewal every 5 min. Few studies have included the carbon dioxide (CO_2_) concentration variable as an indicator of space ventilation. In the few reports where CO_2_ was measured, the concentration was less than 400 ppm [49,59], suggesting that the presence of RNA in aerosols may be underestimated. A SARS-CoV-2-loaded bioaerosols emission rate reduction has been described a few days after the onset of symptoms, which should be investigated to understand the dynamics of the spread of the disease. In this work, the risk of contagion from aerosols in 8 patients is evaluated and compared with transmission by close contact in 19 patients.

## 2. Materials and Methods

### 2.1. Patients Included in the Study

In total, 22 COVID-19-positive patients were included in different study groups. In Group A, 5 volunteer patients were included in environmental sampling (aerosols) and close contact (coughs). In Group B, 14 volunteer patients were enrolled in close contact sampling. In Group C, 3 volunteers were willing to carry out the aerosol sampling. As shown in Table 1, 2/5 patients in Group A were hospitalized, while the rest were in isolation at home. From Group C, no patient was hospitalized, while from Group B, all were hospitalized. The admitted patients were in the Infectious Diseases area or in the Surgery area of the Hospital Clínico Universitario Lozano Blesa (Zaragoza, Spain). Patients presenting severe symptoms were excluded. The mean age was 44.2 ± 20.4 years in Group A, 73.6 ± 13.3 years in Group B, and 28.3 ± 18.3 years in Group C. Vaccination status with COVID-19 mRNA vaccines (Pfizer-BioNTech or Moderna) is indicated in Table 1.

### 2.2. Nasopharyngeal Exudate

Nasopharyngeal swab was standardized to obtain comparable results. Conventional swabs with virus transport medium (VTM) were used. The swabs were introduced 3.5–4.5 cm up one nostril of the patient and rotated 180° five times. Immediately afterward, they were placed in the transport medium and stored in a −17 °C freezer for less than five days.

### 2.3. Air Sampling

A Coriolis µ environmental sampler (Bertin Instruments, Rockville, MD, USA) was used for air sampling. The particle collection efficiency is very efficient for sizes greater than 500 nm and 50% for sizes less than 500 nm. As shown in Figure 2, the aspirated flow is collected in a buffer solution inside a sterile sampling cone, forming a vortex. The particles and microorganisms are centrifuged on the cone wall and are separated from the air. Typically, a flow rate of 300 L/min was used for 10 min in 3 mL of PBS (3000 L of air: 1000 L/mL). However, to maximize the detection limits of viral RNA, we decided to sample for 30–110 min in some experiments. These samplings were performed at 300 L/min, maintaining a stable amount of 3 mL of PBS (adding solution as it evaporated between 10-min periods). The volunteers were asked to perform regular breathing and speech actions. This was suggested to collect aerosols from different respiratory activities. All samples were stored in a freezer at −17 °C for later analysis in a period not exceeding 5 days. Patients did not use masks during the sampling period.

Metabolic CO_2_ levels were measured using Aranet4 Pro meters (Aranet, Riga, Latvia). These devices are designed with a Non-Dispersive Infrared Detector (N-DIR) meter and record measurements with an accuracy of ±50 ppm. The meters were placed next to the Coriolis to determine the CO_2_ concentration at sampling.

The set-up of the experiments varied depending on whether the patient was hospitalized or quarantined at home. In both cases, the equipment was placed 1.5 m from the patient. It was only placed less than 0.5 m in the case of Patient 3 and Patient 4 to perform additional tests, as detailed in the Results section. The air sampler was located at a distance of at least 1 m from the ground (typically 1.3–1.4 m). As shown in Figure 3, the sampler was placed on mobile tables in the hospital (Figure 3a) and on tables in private homes for quarantined individuals (Figure 3b). This allowed us to position the Coriolis at the desired distance from the patient. The CO_2_ m was placed right next to the Coriolis air inlet to measure the CO_2_ level of the collected air. At home, patients were placed in small rooms (>30 m^2^) and doors and windows were closed to prevent the escape of aerosols. However, they were not completely sealed. Forced ventilation (air conditioning systems) could not be closed in hospitals since it could be dangerous for the rest of the patients and health workers. Patients at home performed the experiments in sitting position, while in the hospital they reclined in bed, positioned with around 135° between legs and abdomen.

### 2.4. Cough Sampling

In cough samples collection, volunteers were asked to cough three times to clear their throats and, immediately afterward, cough five times into a 5 cm diameter cone containing 1 mL of PBS. The cones were placed 5 cm from the emitter’s mouth to simulate close interpersonal distance. To homogenize the samples, they were vortexed for 30 s. Samples were stored in a −17 °C freezer and were not kept for more than five days.

### 2.5. Viral RNA Extraction from Masks

Patients who kept masks from previous days donated them for the study. In some patients, this was not possible since they did not use masks during the sampling period. The masks were chopped and suspended in a 50 mL Falcon tube with 10 mL PBS. The tubes were sonicated in ultrasound for 30 s to favor RNA extraction from the mask’s surface. Liquid samples were stored in a −17 °C freezer and analyzed within five days.

### 2.6. RT-qPCR Analysis

All the samples were processed in a biological safety cabinet, complying with the applicable biosafety requirements.

#### 2.6.1. Processing of Environmental Samples

The total sample volume was deposited in Falcon tubes with a 10 kDa Amicon Millipore filter (Millipore, Burlington, MA, USA). They were centrifuged for 15 min at 4000 RPM. Then, 600 µL of the concentrate were transferred to a 600 µL tube of lysis buffer (MagMax Lysis solution, Thermo Fisher, Waltham, MA, USA), homogenized and 200 µL were used for RNA extraction.

#### 2.6.2. Processing of Swabs Samples

The swabs were immersed in 600 µL of lysis buffer (MagMax Lysis solution, Thermo Fisher), and the biological material was detached with rotary movements 3–5 times. They were homogenized and 200 µL was used for RNA extraction.

#### 2.6.3. Nucleic Acid Extraction

According to the manufacturer’s instructions, RNA extraction was performed with the MagMax Core RNA/DNA kit (Thermo Fisher) and the KingFisher Flex System automatic extraction kit (Thermo Fisher). The SARS-CoV-2 PCR used 5 µL of extracted RNA and was performed with the validated assay [80] using two key targets (pan-SARS ESAR and SARS-CoV-2 IP4) in a QuantStudio 5 thermocycler (Applied Biosystems, Waltham, MA, USA) [81].

### 2.7. Identification of SARS-CoV-2 Variants of Concern by Partial Sequencing of the Spike Gene

A pair of primers, F21585 (5′TGCCACTAGTCTCTAGTCAG 3′) and R22341.

(5′GCTGTCCAACCTGAAGAA 3′), were designed for the sequencing of the 5' region of the Spike gene that contains the main mutations that characterize the main VOCs of SARS-CoV-2. For this, the reference sequences of the Wuhan-Hu.1 strain (NC_045512), as well as its variants B.1.1.7 (MZ344997), B.1.617.2 (MZ359841), BA.1 (OL672836), and BA.2 (OM296922) were aligned and compared using multiple alignment software MAFFT version 7.

F21585 and R22341 have full homology with all the VOCs studied and generated an amplification product of 756 nucleotides in Wuhan-Hu.1 (NC_045512). The PCR protocol used was 15 min at 45 °C, 5 min at 95 °C, followed by 40 cycles of 30 s at 95 °C, 1 min at 60 °C, and 1 min at 72 °C, ending with a step of 7 min of extension at 72 °C. The PCR product was purified and sequenced using the Sanger technique (StabVida, Caparica, Portugal). The obtained sequences were compared with those cited above as reference sequences of the main VOCs using Clustal Omega v.1.2.2 software (RRID: SCR_001591).

### 2.8. Determination of the Risk of Infection

Wells and Riley [82] defined the probability of contagion (*P*) in a susceptible individual according to Equation (1), where *n* refers to the inhaled amount of the virus infectious doses (expressed in quanta). The SARS-CoV-2 quantum can be described as the probability of infection of 1 −1/e (63%), although it depends on other factors, such as the immunization of the individual, as expressed by Peng and Jiménez [83].
(1)$$P=1-e^{-n}$$

Cortellessa et al. [84] adapted this same model to calculate the probability of infection (*P*) from the RNA dose as expressed in Equation (2). According to this Equation, *HID*_63_ represents the number of RNA copies needed to initiate the infection with a probability of 63%, estimated in 7 × 10^2^ RNA [85], as discussed in the Results section.
(2)$$P=1-e^{-\frac{D_{total}\left(cv\right)}{HID_{63}}}\;$$

Given the limitations in accurately reproducing the conditions in the event of infection, 3 different scenarios will be considered in this work. In the case of determining the risk of infection by large droplets, it will be considered: (1) that the susceptible individual inhales 25% of the droplets, (2) that the susceptible individual inhales 50% of the droplets, and (3) that the susceptible individual inhales 100% of the droplets. Under these scenarios, the interpersonal distance is considered to be narrow and evaporation and gravitational phenomena are neglected to facilitate the estimation. On the other hand, to determine the risk of contagion by aerosols, it will be considered: (1) that the individual breathes contaminated air for 1 min, (2) that the individual breathes contaminated air for 10 min, and (3) that the individual breathes contaminated air for 1 h. In these assumptions, a respiratory flow of 15 L/min will be considered and all the viral RNA detected in the aerosol is infectious.

### 2.9. Ethical Approval

This study has the approval of the Aragon Community Research Ethics Committee (CEIC Aragón) under references PI20/374 and PI22/130.

## 3. Results and Discussion

### 3.1. Ct Value Should Not Be Taken as a Predictor of Infectiousness

Currently, the gold standard for the diagnosis of SARS-CoV-2 infection is the detection of RNA by RT-qPCR, which has the ability to detect target nucleic acids (<100 copies/mL) with remarkable sensitivity [86,87]. The sensitivity varies depending on the stage of the disease the patient is at. The test’s sensitivity has been estimated at 33% 4 days after contact with the infected person, 62% on the day of onset of symptoms, and 80% 3 days after onset of symptoms [88,89]. The collection technique, time since exposure, and anatomic location of sampling affect the false-negative rate and efficiency of sampling. Bronchoalveolar lavages have the highest sensitivity (93%) compared to samples taken from the upper respiratory tract, followed by sputum samples (72%), nasal swabs (63%), and throat swabs (32%) [88]. The reference method is nasal exudate due to its accessible collection. Currently, whether the viral load estimate obtained by the Ct value (Cycle Threshold in qPCR analysis) is a determining factor for the outcome of the disease is being discussed [90,91,92,93]. However, the quantification of viral RNA is subject to aspects, such as the amount of sample taken, the RT-qPCR kit used, the target used, the thermocycler efficiency, or the storage method [94,95]. For example, RT-qPCR tests that detect more genes have been shown to report lower Ct values [96]. Therefore, individualized interpretation of the test result is necessary. In this work, typified sampling (Section 2.2) has been carried out to reduce the uncertainty between samplings. The mean Ct value was 25.7 ± 7.5 in Group B and 25.7 ± 4.2 in Group C.

A higher concentration of bioaerosols is generated during voluntary coughing, concerning breathing and speech [11]. Cough flows have been observed to be multiphase turbulent clouds with suspended droplets of various sizes [97,98], covering a spectrum from a few nanometers to more than 100 µm [11,99]. Thus, these violent breathing events are critical to infectious diseases spreading [97,100], especially if the pathogens reside predominantly in the upper respiratory tract [101]. Viklund et al. [102] found a higher positivity rate in cough samples (32%) compared to normal breathing (16%). Thus, studying coughing instead of other respiratory droplets is more efficient for detecting infectious individuals. Viral RNA could only be detected in the coughs of 10.5% (2/19). The two patients who showed detectable viral load in coughing (Patient 4 and Patient 8) presented a Ct of 23.16 and 13.48, respectively. Patient 4 presented a complete vaccination schedule, but without a booster dose (2/3), while Patient 8 was fully vaccinated (3/3). Patient 8′s hospitalization made it difficult to follow up due to health problems. However, Patient 4 was able to be sampled on subsequent days, although she only showed positive cough results on the first day of sampling (Table 2). The rest of the patients (17) were negative in coughs, despite presenting lower Ct values.

The viral load found in the cough samples was 7.3 × 10^5^–9.1 × 10^5^ copies/mL (Ct 28.5–27.4) and 8.7 × 10^8^–6.7 × 10^8^ (Ct 18.2–18.2) in Patients 4 and 8, respectively. These values are higher than Viklund et al. [102] (Ct 29.5–36.5) for patients with Ct between 17.2–26.4. However, Patient 8 showed a higher viral load, although it cannot be compared due to a much lower Ct in the nasopharyngeal exudate.

A correlation between Ct and the infectivity of individuals has been previously reported [103,104]. This indicator has even been taken as an epidemiological prediction tool [105]. However, a clear relationship between patient infectiousness and viral load has not been found in this work. Patients 6, 7, and 19, with Ct values of 16.7, 16.8, and 15.8, were negative for cough samples, while Patient 4 with a Ct of 23.2 emitted SARS-CoV-2-loaded droplets under the same terms. Another case of interest is Patient 5, with characteristics similar to Patient 4, in terms of age and viral load. As seen in Table 3, no viral RNA was detected in either the patient’s oropharyngeal exudate or cough samples.

Patients 1, 2, and 5 wore masks the days before sampling. These masks were analyzed by PCR for SARS-CoV-2 detection. The only positive sample was obtained from Patient 1, who presented a Ct of 28.1 at sampling. He was negative on cough and air, although the surgical mask he had worn for the previous three days was positive, with an average viral load of 29.5 (3.7 × 10^3^ copies/mL). These results indicate that Patient 1 was infectious on previous days.

### 3.2. Detection of SARS-CoV-2-Laden Bioaerosols

The average Ct value was 24.6 ± 4.3 in Group A and 25.7 ± 4.2 in Group C. A total of 88 air samples from eight different patients from group A and C were collected. All patients included in this subgroup were infected with the variant of concern (VOC), Omicron BA.1 (Appendix A). Only seven air samples (8.0%) were detectable for SARS-CoV-2 RNA levels and another eight samples were suspicious (9.0%; Ct > 38). As seen in Table 4, the 15 positive/suspicious samples were collected from the same patient (Patient 20), although on the second day, the samples were taken in the company of two other patients (Patient 21 and Patient 22), since they constituted a coexisting unit.

In the samples collected on the first day of the trial (2 days after the onset of symptoms), positivity was found in 68.8% (11/16) of the samples, with three suspicious samples. However, on the second day of the trial (4 days after symptom onset), suspicion was only found in 14.8% of samples (4/27). The aerosol viral load results (1.1–4.8 copies/m^3^) were consistent with those previously reported in the literature [24,31,32,33,34,35,36,37,38,39,40,41,42,43,44,45,46,47,48,49,50,51,52,53]. Given the airborne persistence of SARS-CoV-2, it is important to consider that viral RNA may have been collected from inactivated viruses. Therefore, they would not have the ability to infect humans. Since it is not possible to make precise estimates for the percentage of viable virus, the maximum possible persistent viral load will be considered in this work.

The global acceptance of the COVID-19 airborne spread allowed for an improvement in the preventive strategies, including new techniques for epidemiological management, such as the measurement of exhaled CO_2_ as an indicator of the air renewal and, consequently, the risk of contagion [35,83]. The presence of viral aerosols in some infected individuals makes it necessary to implement air control measures in collective environments.

Even under poor ventilation conditions, the remaining patients did not shed detectable viral RNA. Patient 3 (Table 5), with a Ct of 19.1, achieved 2400 ppm metabolic CO_2_, implying about ~5% air coming from the individual. However, this same patient tested negative for cough samples on days 2, 4, and 6 after symptom onset (Ct 19.1, Ct 30.6, and Ct 32.6, respectively), despite presenting viral load in the oropharyngeal swab on day 2 (Ct 21.2).

Patient 4, with a Ct of 24.6 (day 2 after the onset of symptoms), showed positive coughs and positive oropharyngeal exudate (Ct 28.9). However, the aerosol samples were negative, even in a unique sampling where they were concentrated at 33,000 L of air in 2 mL PBS, and the patient was kept talking and breathing at 50 cm distance from the Coriolis. As a control, the collection cone wall and the air inlet were sampled with a swab, which also tested negative for SARS-CoV-2. Similar tests were performed on Patient 3, where he used the Coriolis at less than 20 cm and sang for 10 min, but viral RNA was not detected.

In addition to rapid rates of loss of infectivity [79], the airborne viral load of SARS-CoV-2 is reduced. Although the published studies cannot be directly compared due to the variation in conditions, the reported concentrations point to a viral load between 5.0 and 11.9 × 10^3^ copies/m^3^ of air (Appendix A), and an equivalent mean concentration of 3.1 ± 2.9 copies/L of air is deduced from a total of 3085 samples (313 positives; 10.2%). However, it is necessary to consider that the values should be corrected according to the efficiency of each sampling device. Thus, our results for aerosolized viral load (1.1–4.8 copies/m^3^) in samples collected from Patient 4 were consistent with those previously reported.

### 3.3. Do Superspreaders Predominate in the Emission of Bioaerosols with SARS-CoV-2?

Uncertainty persists about the interrelationship between the exposure environment and the transmission networks of the SARS-CoV-2 virus. According to the CDC, droplet transmission (close contact) is predominant, while fomites and aerosols explain special propagation events in punctual events [106]. The role of individuals in the asymptomatic phase has accounted for a substantial portion of infections (Appendix A) [107,108,109,110,111]. It has been estimated that 44% (95% CI; 30–57%) of secondary cases became infected during the incubation period [112]. The heterogeneity of infectious disease transmission is a well-known concept that has been studied in various epidemic scenarios. Woolhouse et al. [113] identified a statistical pattern, known as the 20/80 rule, which indicates that 80% of new infections are associated with only 20% of those infected. According to this concept, there is a central high-risk group that may be associated with the massive expansion of the infectious disease [114,115,116]. In the case of SARS-CoV-2, the dynamics of the spreads pointed to similar conclusions [117,118]. Even the central high-risk group has been reduced to 10–21% of cases [118,119]. Heterogeneity in transmission has already been observed in the dynamics of other coronaviruses, such as SARS-CoV [117,120,121]. Moreover, other respiratory viruses with similar viral characteristics, such as H1N1, do not exhibit these patterns of spread [122].

Sustained superspreading events could explain the massive infection of individuals [123,124]. However, it may also be related to individuals’ environmental, behavioral and social factors that influence the dynamics of virus transmission, which could depend on the configuration of specific outbreaks [125]. Beldomenico [126] proposed that the superspreaders might not be random and depend on other superspreaders. Then, superspreaders’ secondary infections are more likely to rise to new superspreaders. Although the origin of superspreading is unknown in detail, it is mainly attributed to physiological issues of the individual, making the infectiousness of the individual challenging to predict [127]. A study by Edward et al. [128], on the aerosol emission rate of 200 healthy individuals, emphasized that biological differences could affect virus transmission. The work reported that 20% of the study participants accounted for 80% of the aerosols emitted.

A substantial number of reported COVID-19 cases are from superspreader events, where secondary cases are disproportionately higher than expected based on basic reproduction number (R_0_) [129]. In this study, of the four infected individuals who were quarantined, only two had a viral load in coughs and only one in aerosols emitted. The nuclear families of the three individuals who were not infectious through aerosols did not become infected, while Patient 4 infected both her family (*n* = 3) and her partner’s family (*n* = 4). Our results point to reduced contagiousness in the incubation period and a predominant role of superspreaders when the disease is already established.

Although SARS-CoV-2 can be explained by the high viral spread of superspreaders [126], it remains unknown what determines individual overdispersion in transmissibility, so it is not possible to discriminate between superspreaders and non-spreaders [115]. Then, a broad understanding of the role of individual variation in the ecology of respiratory viruses in controlling infection transmission is still required.

### 3.4. Probable Time Required to Inhale SARS-CoV-2 Bioaerosols to Undergo an Infection

The information on the infective doses sufficient to cause infections with SARS-CoV-2 is limited and subject to the study model. It is known that viral load plays a relevant role in viral kinetics in aerial pathogen transmission [130]. Fain et al. [131] found that high initial inocula lead to brief infections but with higher peak viral titers (10^6^); smaller initial inocula (10^1^) reduce the peak viral titer but make the infection last longer. The estimated dose-infection effect for humans has been determined in vitro. Zhang et al. [132] mathematically determined an infective constant (k) of 6.4 × 10^4^ to 9.8 × 10^5^ copies for the onset of infection. In contrast, Therese et al. [133] reported a minimum dose of 4.0 × 10^4^ copies for an in vitro infection in Vero B4 cells. Only 21% of the 109 SARS-CoV-2 samples isolated from patients were infectious enough to initiate a new infection. A recent study by Killingley et al. [134] studied the course of infection in human volunteers. Thirty-six volunteers were intranasally infected with 10 TCID_50_ (SARS-CoV-2/human/GBR/484861/2020), reaching peak RNA levels of 5 dpi (~8.9 log10 copies/mL) and infecting 18/36.

Other studies with the same objective have been carried out in animal models (Appendix A). For aerosols, the median aerosol infectious dose tested in small animals (hACE2 mice) is 630 copies (infection rate 2/2) [135], while in large animals (African Green Monkeys), it is 2.0 × 10^3^ copies (infection rate 2/2) [107]. Infections have been reported from 500 copies (infection rate 1/6 in ferrets) [136] to 3 × 10^6^ copies (8/8 African Green Monkeys) [137] by intranasal inoculations. Some cases have been reported where a lower attack rate has been obtained with higher viral loads; for example, inoculations of 7.0 × 10^4^ copies in mice (hACE2) infected only 7/19 mice [138] and 7/9 Syrian hamsters [139], while in ferrets, an attack of 6/6 ferrets with a dose of 5.0 × 10^4^ copies [136].

According to the literature (Appendix A), the median infective dose demonstrated was 1.0 × 10^3^ copies to infect 100% of the animals included in the study [135], while Gale et al. [85] estimated 7.0 × 10^2^ copies to infect 63% of human individuals (*HID*_63_). However, the infection dynamics are variable from individual to individual, and probabilistic adjustments are limited [140]. By working out the dynamics of the infection in probabilistic terms, it is possible to find infections caused by a single virus, although the probability is minimal [141].

Considering an average persistent viral load of 3.1 ± 2.9 copies/L and a homogeneous concentration of viral particles in the environment and a *HID*_63_ of 7.0 × 10^2^ RNA copies, if we assume that 100% of the viruses are viable, breathing 15 L of air (1 min), the probability of infection (*P*) is 0.68, while it reaches 0.99 and 1.0 after inhaling 150 (10 min) and 900 L (1 h). According to this model, a contagion probability of 1.0 would be reached after 15 min of breathing. However, it would be necessary to apply several correction factors that include, for example, the percentage of viruses that do not persist (and, therefore, are not infectious) or the probability of inhalation of these contaminated aerosols.

### 3.5. Viral Spreading Is Heterogeneous

Assuming an average viral load of 8.2 × 10^5^ copies/mL found in the cough samples of this work, there is a concentration (*c_p_*) of 4.9 × 10^6^ ± 6.1 × 10^6^ aerosols per cough in individuals infected with respiratory viruses [142], with an average aerodynamic radius (*r*) of 0.4 μm [11]. Given the variability between individuals [112], the maximum deviation equates the spread by aerosols (=1.1 × 10^7^). Under these assumptions, in each cough, a maximum liquid volume of aerosols (*v*) of 2.95 × 10^−4^ mL is considered, according to Equation (3).
(3)$$v=\frac{4}{3}\;\pi \;r^{3}c_{p}\;,$$

Note that for this estimate, a perfect sphere has been considered instead of an irregular droplet, an average aerodynamic diameter of 0.8 μm (rather than a distribution of particle sizes), and an average amount of aerosols disseminated in coughs in patients infected with H1N1.

Following this approach, Patient 4 was able to shed 242 copies of RNA in his cough, while Patient 8 was able to shed around 115,050 copies. If 100% of cough droplets had been inhaled, the probability of infection would be 0.98 in Patient 4′s cough, while in Patient 8, it would be 1.0. In the first case, the viral load resulting from coughing would be equivalent to breathing air for around 10 min at a flow of 15 L/min, with an assumed homogeneous concentration of 3.1 ± 2.9 copies/L. However, in the second case, it would be equivalent to breathing air loaded with SARS-CoV-2 for a minimum of 41 h.

## 4. Conclusions

The spread patterns of SARS (2003) could not be explained by conventional epidemic models, which assumed homogeneity of transmission. In the same way, SARS-CoV-2 has shown a propagation dynamic that is difficult to parameterize. This variability in the viral spread between people could explain the different growth rates of infected people between populations, although the cause of this variability is still unknown.

In this work, it has been found that the Ct value cannot be considered as an indicator of the individual’s infectiousness, since it has not been possible to correlate with the viral load disseminated in aerosols and coughs. Given the limited literature in this regard and the variability in sampling and diagnostic equipment, this parameter should be reconsidered in matters of epidemiological prediction. In our study, patients with a very low Ct (>15) shed less (or no) viral load, while patients with a Ct of up to 27 generated high viral concentrations in the environment.

The viral load in coughs was different between the samples of the two individuals (*n* = 19) that were positive. Concentrations between 7.3 × 10^5^ to 8.7 × 10^8^ copies/mL were found, implying an approximate viral load of between 1.1 × 10^2^ and 1.0 × 10^5^ copies/cough. In this subgroup, the mean Ct was 25.7 ± 7.0, and the patients who disseminated viral load had a Ct of 23.2 and 13.5. In this case, the patient with the lowest Ct (13.5) had the highest viral load spread in coughs (8.7 × 10^8^ copies/mL). However, not all patients with a positive Ct in oropharyngeal samples coughed up the viral load. In one of the patients with positive sampling, it was possible to carry out a follow-up, where a loss of infectiousness was observed from the third day after the onset of symptoms.

Only one patient shed detectable viral RNA in the air. Additionally, he was the only one who infected seven other individuals. Of the rest of the patients, there is no evidence of secondary infections. The viral load found in the air (1.1–4.8 copies/m^3^) was consistent with that previously reported in the literature. It was considerably reduced from the first to the third day after symptom onset. In patients with high rates of viral shedding, a cough can be equivalent to 10 min of breathing, making more necessary to improve ventilation and air purification strategies in shared indoor spaces.

It is still necessary to understand the role of the different routes of spread in COVID-19 spreading, considering the influence of parameters, such as the vaccinated population rate and their immunity or the predominant variants. Specifically, knowledge about viral dynamics in aerosols and superspreading events is essential for understanding the spread heterogeneity and better managing future pandemics. Since a low SARS-CoV-2 virus load can initiate an infection, the task of designing epidemiological prediction models is complex, despite being necessary to manage infectious diseases more efficiently in the future.

## Figures and Tables

**Figure 1 jcm-11-02607-f001:**
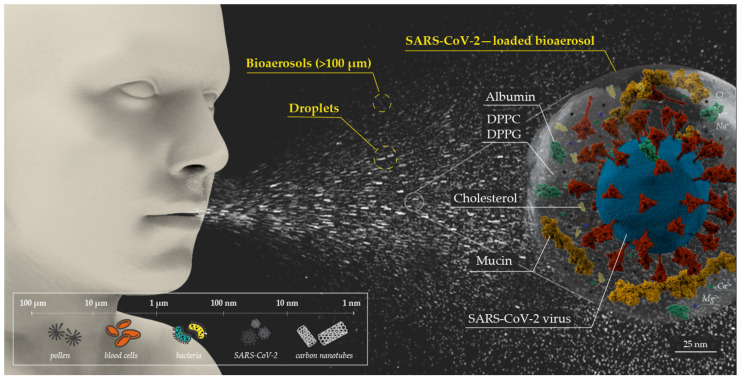
Schematic representation of bioaerosols and droplets emission. Where, DPPC is dipalmitoylphosphatidylcholine protein, and DPPG is dipalmitoylphosphatidylglycerol protein.

**Figure 2 jcm-11-02607-f002:**
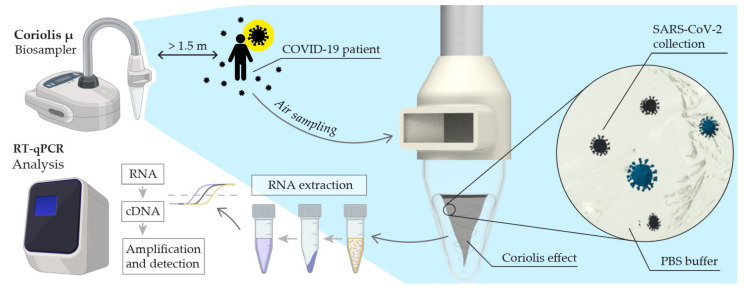
Schematic representation of air sampling and analysis method. Where, PBS is Phosphate Buffered Saline.

**Figure 3 jcm-11-02607-f003:**
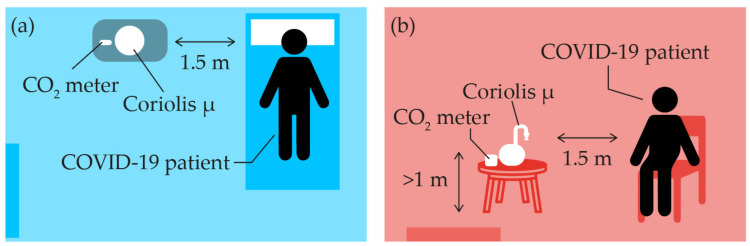
Schematic representation of air sampling set-up at (**a**) the hospital and (**b**) in private homes.

**Table 1 jcm-11-02607-t001:** Characteristics of the included patients.

Patient	Age	Gender	DASO *	Sampling Period	Initial Ct	Final Ct	Vaccines	Hospitalization	Group
Patient 1	59	Male	3 days	1 day	28.1	N/A	3	Yes	A
Patient 2	45	Female	2 days	1 day	29.8	N/A	3	Yes	A
Patient 3	59	Male	2 days	3 days	19.3	30.4	3 ^1^	No	A
Patient 4	22	Male	1 day	4 days	23.7	36.1	2	No	A
Patient 5	26	Male	1 day	3 days	23.2	29.3	2	No	A
Patient 6	89	Male	0 days **^†^**	1 day	16.7	N/A	3	Yes	B
Patient 7	75	Male	0 days **^†^**	1 day	16.8	N/A	3	Yes	B
Patient 8	61	Male	0 days **^†^**	1 day	13.5	N/A	3	Yes	B
Patient 9	59	Female	1 days **^†^**	1 day	32.7	N/A	No	Yes	B
Patient 10	84	Male	2 days **^†^**	1 day	32.8	N/A	3	Yes	B
Patient 11	63	Male	0 days **^†^**	1 day	33.1	N/A	3	Yes	B
Patient 12	89	Male	0 days **^†^**	1 day	28.8	N/A	3	Yes	B
Patient 13	68	Male	1 day **^†^**	1 day	24.8	N/A	3	Yes	B
Patient 14	69	Female	6 days **^†^**	1 day	26.1	N/A	No	Yes	B
Patient 15	88	Female	1 day **^†^**	1 day	34.8	N/A	3	Yes	B
Patient 16	93	Male	1 day **^†^**	1 day	28.8	N/A	3	Yes	B
Patient 17	88	Male	1 day **^†^**	1 day	33.3	N/A	3	Yes	B
Patient 18	54	Female	1 day **^†^**	1 day	21.8	N/A	3	Yes	B
Patient 19	67	Male	7 days **^††^**	1 day	15.8	N/A	3	Yes	B
Patient 20	22	Female	2 days	2 days	27.1	29.2	2	No	C
Patient 21	49	Female	2 days	1 day	29.3	N/A	2	No	C
Patient 22	14	Female	2 days	1 day	21.2	N/A	2	No	C

* DASO: Days after symptoms onset; ^1^ The patient inadvertently received the vaccine while infected with the COVID-19 disease; **^†^** The period refers from the diagnosis; **^††^** The patients were included because they were not vaccinated, despite being diagnosed more than 48 h earlier.

**Table 2 jcm-11-02607-t002:** Patient 4 follow-up samples. Where, pan-SARS ESAR refers to Sarbeco E gen detection, and SARS-CoV-2 IP4 refers to RdRp gen detection.

Sample	Amplification	Day 2	Day 3	Day 4	Day 6	Day 7
Nasopharynx	pan-SARS ESAR	25.5 (5.5 × 10^6^ copies/mL)	23.8 (2.0 × 10^7^ copies/mL)	25.9 (4.8 × 10^6^ copies/mL)	33.8 (2.3 × 10^4^ copies/mL)	36.1 (4.9 × 10^3^ copies/mL)
	SARS-CoV-2 IP4	24.6 (3.2 × 10^6^ copies/mL)	23.1 (2.5 × 10^7^ copies/mL)	25.4 (5.4 × 10^6^ copies/mL)	34.4 (1.2 × 10^4^ copies/mL)	Not detected
Oropharyngea	pan-SARS ESAR	29.1 (4.8 × 10^5^ copies/mL)	29.3 (4.8 × 10^5^ copies/mL)	Not detected	Not detected	Not detected
	SARS-CoV-2 IP4	28.9 (3.2 × 10^5^ copies/mL)	29.4 (3.7 × 10^5^ copies/mL)	Not detected	Not detected	Not detected
Coughs	pan-SARS ESAR	28.5 (7.3 × 10^5^ copies/mL)	Not detected	Not detected	Not detected	Not detected
	SARS-CoV-2 IP4	27.4 (9.1 × 10^5^ copies/mL)	Not detected	Not detected	Not detected	Not detected

**Table 3 jcm-11-02607-t003:** Patient 5 follow-up samples. Where, pan-SARS ESAR refers to Sarbeco E gen detection, and SARS-CoV-2 IP4 refers to RdRp gen detection.

Sample	Amplification	Day 2	Day 3	Day 4
Nasopharynx	pan-SARS ESAR	25.1 (7.8 × 10^6^ copies/mL)	23.3 (2.7 × 10^7^ copies/mL)	28.7 (6.7 × 10^5^ copies/mL)
	SARS-CoV-2 IP4	24.6 (8.9 × 10^6^ copies/mL)	22.9 (2.8 × 10^7^ copies/mL)	28.3 (7.4 × 10^5^ copies/mL)
Oropharyngeal	pan-SARS ESAR	Not detected	Not detected	Not detected
	SARS-CoV-2 IP4	Not detected	Not detected	Not detected
Coughs	pan-SARS ESAR	Not detected	Not detected	Not detected
	SARS-CoV-2 IP4	Not detected	Not detected	Not detected

**Table 4 jcm-11-02607-t004:** Air samples where viral RNA has been detected. Where, pan-SARS ESAR refers to Sarbeco E gen detection, and SARS-CoV-2 IP4 refers to RdRp gen detection.

Sample	CO_2_ Levels	Amplification	Patient 20
			Day 1
1	500–1000 ppm	pan-SARS ESAR	36.7 (1.1 × 10^3^ copies/mL)
		SARS-CoV-2 IP4	Not detected
2	500–1000 ppm	pan-SARS ESAR	36.6 (2.3 × 10^3^ copies/mL)
		SARS-CoV-2 IP4	36.8 (2.0 × 10^3^ copies/mL)
3	500–1000 ppm	pan-SARS ESAR	37.2 (1.6 × 10^3^ copies/mL)
		SARS-CoV-2 IP4	Not detected
4	500–1000 ppm	pan-SARS ESAR	38.9 (Suspicious)
		SARS-CoV-2 IP4	38.8 (Suspicious)
5	500–1000 ppm	pan-SARS ESAR	Not detected
		SARS-CoV-2 IP4	38.7 (Suspicious)
6	500–1000 ppm	pan-SARS ESAR	37.9 (9.6 × 10^2^ copies/mL)
		SARS-CoV-2 IP4	Not detected
7	1000–1500 ppm	pan-SARS ESAR	37.2 (1.5 × 10^3^ copies/mL)
		SARS-CoV-2 IP4	39.2 (Suspicious)
8	1000–1500 ppm	pan-SARS ESAR	38.8 (Suspicious)
		SARS-CoV-2 IP4	38.3 (Suspicious)
9	1000–1500 ppm	pan-SARS ESAR	39.8 (Suspicious)
		SARS-CoV-2 IP4	Not detected
10	1500–2000 ppm	pan-SARS ESAR	35.5 (4.8 × 10^3^ copies/mL)
		SARS-CoV-2 IP4	36.3 (Suspicious)
11	1500–2000 ppm	pan-SARS ESAR	37.0 (1.8 × 10^3^ copies/mL)
		SARS-CoV-2 IP4	38.3 (Suspicious)
			**Patient 20–22**
			**Day 3**
12	500–1000 ppm	pan-SARS ESAR	39.0 (Suspicious)
		SARS-CoV-2 IP4	Not detected
13	1000–1500 ppm	pan-SARS ESAR	Not detected
		SARS-CoV-2 IP4	38.9 (Suspicious)
14	1500–2000 ppm	pan-SARS ESAR	39.0 (Suspicious)
		SARS-CoV-2 IP4	Not detected
15	1500–2000 ppm	pan-SARS ESAR	39.5 (Suspicious)
		SARS-CoV-2 IP4	Not detected

**Table 5 jcm-11-02607-t005:** Patient 3 follow-up samples. Where, pan-SARS ESAR refers to Sarbeco E gen detection, and SARS-CoV-2 IP4 refers to RdRp gen detection.

Sample	Amplification	Day 2	Day 4	Day 6
Nasopharynx	pan-SARS ESAR	19.9 (1.3 × 10^7^ copies/mL)	30.8 (1.4 × 10^4^ copies/mL)	33.4 (2.5 × 10^4^ copies/mL)
	SARS-CoV-2 IP4	19.1 (2.5 × 10^7^ copies/mL)	30.6 (1.5 × 10^4^ copies/mL)	32.6 (3.7 × 10^4^ copies/mL)
Oropharyngeal	pan-SARS ESAR	21.9 (3.7 × 10^6^ copies/mL)	Not detected	38.9 (Suspicious)
	SARS-CoV-2 IP4	21.2 (5.9 × 10^6^ copies/mL)	Not detected	37.2 (1.6 × 10^3^ copies/mL)
Coughs	pan-SARS ESAR	Not detected	Not detected	Not detected
	SARS-CoV-2 IP4	Not detected	Not detected	Not detected

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
