# Peer review of "SARS-CoV-2 Droplet and Airborne Transmission Heterogeneity"

_jcm, 2022, doi:10.3390/jcm11092607_

Round 1

Reviewer 1 Report

The subject of this paper is very interesting since its target was to compare airborne viral load between different patients positive to covid-19 by direct air sampling in order to determine if airborne transmission is mainly due to a few infected people, called super-spreaders.

Three types of testing were performed using RT-PCR measurements of samples:

  • Nasopharyngeal swab analysis which is a standard way to detect positivity to CoVID-19.
  • Cough samples
  • Airborne samples acquired with a Bertin Coriolis instrument dedicated to aerosol collection.

Despite the interest of the subject my opinion is that the paper is missing its target.  First of all, I found this paper difficult to read and therefore to assess, but scientific reasons support my recommendation, some of which are exposed below:

My major concern is that the authors do not seem to know, or at least do not discuss, the difference between airborne transmission and the one due to droplets of larger size that behave in a ballistic way. “Airborne” should be reserved to these microdroplets which have a size, either at emission or after evaporation, allowing them to linger in the air for very long time. The problem of evaporation itself should deserve a more complete discussion.   Coughing in a cone, as described in the paper, although the distance to the emitter and the size of the cone is not mentioned, could collect large particles together with the airborne ones. Therefore, the part of the study related to “coughing” is in the present form meaningless for the purpose of the paper, unless if the distance emitter/cone is large.  In the same way air sampling with the Coriolis instrument is either .5 or 1.5 meter, but I found no indication of the distance that what used when RT-PCR tests was positive for the air sampling. Moreover, only air sampling has led to negative results excepted for one patient. I recommend to the authors to read the following paper: “Close proximity risk assessment for SARS-CoV-2 infection” G. Cortellessa et al, Science of the Total Environment 794 (2021) 148749, which is not cited in the list of their references.

Another important concern is the question of the definition of the dose in the airborne transmission which is not well defined in this paper although it is just the (viral -or quantum- density in the air) x (pulmonary inhalation flow rate) x (time of exposure).  Then the question is the risk assessment is not well examined. It would deserve a discussion of the dose/risk function which leads to a probability of infection. From this point of view the terminology of “Minimum Infective Dose” used by the authors is also very misleading as discussed by Haas et al (Haas CN, Rose JB, Gerba CP. Quantitative Microbial Risk Assessment. Hoboken, NJ: John Wiley & Sons, Inc.; 2014., see also https://doi.org/10.1371/journal.pcbi.1005481) as it suggests some threshold effect although a single pathogen is enough to start an infection (although with a small probability, but everything is statistical in this story). I know that this terminology is widely used but (see Haas) should be replaced by Median or other term instead of Minimum.

The lifetime of the virus in aerosols is also a very important question which is discussed in the paper. However, there are considerable discrepancy in the literature on the subject. With the lifetime reported recently by the Bristol group in a preprint not yet peer-reviewed, and highlighted in the present manuscript , airborne transmission should be negligible, which shed suspicion on their results considering the numerous evidences that it is.

Another question is that RT-PCR detection of viral contents is not an indication of their infectiousness as inactivated viruses can be detected in this way. This should be discussed.

I also found no mention to the seminal work of William Wells on airborne transmission and to his book of 1955. It would deserve to be mentioned in the bibliography.

To finish I know that detection of virus in the air, even inactivated, is a very hard task, therefore the authors should consider submitting again considering the previous observations and staying modest on the conclusion concerning superspreading events.

Reviewer 2 Report

The study is important and well performed, but needs some improvements before acceptable.

First, it is crucial that the authors make a statement which variants were identified. It is obviuos these days that Omicron has a higher infection rate than any other strain before, thus it is likely that even the lowest aerosolic concentrarion may lead to an infection in vivo.

Second, it is important to take into account that coughing produces many large droplets and less small aerosolic virus particles, but the latter are assumed to be more relevant for viral spreading as they are also able to leave leaky masks (either not correctly worn or of bad quality).

Moreover, this reviewer fully agrees that the ct-value does not say anything on infectiousness. infectiousness strongly depends on host factors that are responsible for the susceptibility. In cell cultures it is likely that the novel generations of cell cultures used for the detection of SARS-CoV-2 are much more sensitive than early attempts in 2020 when the pandemics started.

Additionally, an Austrian study has clearly shown that ct values variy among different labs and different techniques, and even the same PCR chemistry performs different if the thermocycler is different. This is not sufficiantly discussed, but is also not part of the study.

Finally, for me as a virologist it would be extremely helpful if the authors could provide a more detailed scheme/figure showing how the experimental setup was performed. I have already participated several workshops with aerosol researchers and learned much from these colleagues, but the waste majority of virologists (and other disciplines) has not. It would strengthen the role of aerosolic research, which in my view is crucial for future research and was underestimated in the current pandemics.

Round 2

Reviewer 1 Report

Following my first review where I have made a lot of remarks the authors have made a great work  in order to take into account my remarks and they have gratly improved their paper which is now much more understandable. Therefore, given the interest of this kind of study I recommend now publication.